# Amyotrophic Lateral Sclerosis and Serum Lipid Level Association: A Systematic Review and Meta-Analytic Study

**DOI:** 10.3390/ijms24108675

**Published:** 2023-05-12

**Authors:** Teresa Pardo-Moreno, Himan Mohamed-Mohamed, Sami Suleiman-Martos, Juan José Ramos-Rodriguez, Antonio Rivas-Dominguez, Lucía Melguizo-Rodríguez, José L. Gómez-Urquiza, Beatriz Bermudez-Pulgarin, Victoria Garcia-Morales

**Affiliations:** 1Department of Physiology, Faculty of Health Sciences—Ceuta, University of Granada, 51001 Ceuta, Spain; terepm11@gmail.com (T.P.-M.);; 2Servicio Andaluz de Salud, 18014 Granada, Spain; 3Department of Cell Biology, Faculty of Biology, University of Seville, 41012 Seville, Spain; 4Department of Nursery, Faculty of Health Sciences—Ceuta, University of Granada, 51001 Ceuta, Spain; 5Physiology Area, Department of Biomedicine, Biotechnology and Public Health, Faculty of Medicine, University of Cádiz, 11003 Cádiz, Spain

**Keywords:** amyotrophic lateral sclerosis, lipids, meta-analysis, cholesterol, neurodegeneration

## Abstract

Amyotrophic lateral sclerosis (ALS) is a fatal neurodegenerative disease with unknown etiology. Many metabolic alterations occur during ALS progress and can be used as a method of pre-diagnostic and early diagnosis. Dyslipidemia is one of the physiological changes observed in numerous ALS patients. The aim of this study is to analyze the possible relationship between the rate of disease progression (functional rating scale (ALS-FRS)) and the plasma lipid levels at the early stage of ALS. A systematic review was carried out in July 2022. The search equation was “Triglycerides AND amyotrophic lateral sclerosis” and its variants. Four meta-analyses were performed. Four studies were included in the meta-analysis. No significant differences were observed between the lipid levels (total cholesterol, triglycerides, HDL cholesterol, and LDL cholesterol) and the ALS-FRS score at the onset of the disease. Although the number of studies included in this research was low, the results of this meta-analytic study suggest that there is no clear relationship between the symptoms observed in ALS patients and the plasma lipid levels. An increase in research, as well as an expansion of the geographical area, would be of interest.

## 1. Introduction

Amyotrophic lateral sclerosis (ALS) is a neurodegenerative disease with a fatal outcome; the disease causes upper and lower motor neuron degeneration, leading to muscle atrophy and paralysis [1]. The cause of ALS is unknown, and it is considered a heterogeneous syndrome [2]. Patients with ALS have a maximum life expectancy of up to 5 years after diagnosis and ultimately die due to respiratory failure. In general, there are two classifications of ALS according to its etiology—95% of patients develop the disease sporadically, and approximately 5% of cases have a family background, the latter being caused by different gene mutations (the C9orf72 gene, Cu/Zn-superoxide dismutase 1 (SOD1) gene, TDP43, and FUS, among others) [3]. There are more than 30 genes described that trigger the onset of ALS, and it is considered today to be a complex disease to diagnose at the early stages. In Europe, the incidence rate is 2–3 cases per 100,000 individuals; so, ALS is considered a rare disease. However, the incidence of ALS is lower in East Asia (0.8 per 100,000 individuals) and South Asia (0.7 per 100,000 individuals). The data on survival are different in various regions: from 24 months in Europe to 48 months in Central Asia [4,5]. The initial symptom presentation of ALS can be spinal-onset disease (muscle weakness of the lower and upper limbs) or bulbar-onset disease (dysarthria (speech deficit) and dysphagia (swallowing deficit)). Moreover, almost 50% of ALS patients present cognitive decline (of whom 13% develop frontotemporal dementia) and/or behavioral alterations during the progress of the disease [6,7,8]. For these reasons, ALS is characterized as a neurodegenerative disease rather than a neuromuscular pathology. Diverse environmental and lifestyle factors have been described in relation to ALS. A higher frequency has been reported in groups of athletes; so, moderate physical activity is a risk factor for ALS. Preliminary studies suggest that exposure to smoking is a risk factor for developing ALS, while high circulating lipid levels, type 2 diabetes, and female oral contraceptive seem to be neuroprotective [9,10].

Several pathological mechanisms have been described as occurring during ALS development, including neuroinflammation, neurotoxicity induced by glutamate, RNA processing and metabolism, axonal transport, vesicle trafficking, oxidative stress, and mitochondrial dysregulation [11]. Approximately 97% of ALS patients present TDP-43 proteinopathy (TDP-43 cytoplasmic aggregates in residual motoneurons). Various types of protein aggregates have been observed in ALS patients. So, these aggregates appear to be neurotoxic and to induce neurodegeneration. On the other hand, other molecular alterations have been reported in ALS animal models, such as aberrant RNA metabolism and DNA repair deficit [12,13]. The excitotoxicity mechanism is the most accepted in ALS pathologies and is a common mechanism in all forms of ALS. Motoneurons are very sensitive to toxicity induced by intracellular calcium following excessive glutamate stimulation by the 3-hydroxy-5-methyl-4-isoxazolepropionic acid (AMPA) receptors. Alterations in glutamate sensitivity on the AMPA receptor have been evidenced in ALS models and ALS patients [14]. In addition, high glutamate concentrations were determined in cerebrospinal fluid due to an excitatory amino acid transporter deficit in astroglial cells, inducing synaptic glutamate abundance and motoneuron toxicity [15,16].

Most of the investigations into the pathophysiological mechanisms of ALS have been in animal models (rat, mouse, zebrafish, fly, and worm) based on gene mutations. These mutations cause motoneuron degeneration and several motor symptoms. However, these models are limited and do not recapitulate human disease. Moreover, only 5% of ALS patients present familial ALS, while 95% present sporadic ALS; therefore, studying ALS in animal models is not representative [5].

The diagnosis of ALS involves a process of clinical studies to exclude other possible pathologies with common symptoms, such as swallowing problems. ALS diagnosis is based on the El Escorial criteria [17]. The diagnosis requires the presentation of progressive weakness and a spreading evolution in relevant regions (bulbar regions, with speech and swallowing problems), cervical regions (upper limb mobility loss), thoracic regions, or lumbar regions (lower limb mobility loss), with evidence of denervation in electromyography. Unfortunately, these symptoms appear when the disease is advanced, making it more difficult to treat the patients. Currently, an affirmative diagnosis in symptomatic individuals is based on blood sample analysis, as well as the evaluation of muscle biopsy samples, muscle function degree, and behavior deficits (speech disturbance, trouble swallowing, and loss of muscle strength). Diverse serum and plasma biomarkers have been proposed as possible candidates to predict disease progression and life expectancy in patients with ALS (serum urate, serum creatinine, serum chloride, and increased serum and cerebrospinal fluid neurofilament levels) [18,19,20,21]. Liquid chromatography–mass spectrometry technology has allowed us to identify plasma metabolite profiles and to determine a regression between metabolites and ALS risk [22,23,24]. Proteomic analysis [25] considers changes in the metabolism of the excitatory neurotransmitter glutamate [26] and increases in the TDP-43 protein [27], as demonstrated by chromatography–mass spectrometry technology in blood samples, serum, or cerebrospinal fluid. ALS patients often have an impaired energy metabolism, including glucose and lipid metabolism [28]. A number of authors have described dyslipidemia as one of the most important milestones in ALS, in addition to an increment in body mass index (BMI) or insulin resistance [29,30,31,32,33,34]. Nevertheless, other authors have shown conclusions that take other directions [30,35,36]. Different research investigations carried out in animal models with genetic ALS (SOD1G93A model) showed lower postprandial lipid levels (triglycerides and total cholesterol) at a pre-symptomatic stage compared to those of a wild-type mouse model [37]. ALS patients with hyperlipidemia are characterized by a low body mass index; in fact, subjects with a high body mass index and progressive weight gain are less prone to suffering from ALS [38], making the concept of increased lipid levels in ALS counterintuitive.

Overall, it has been collectively established that body mass index could be predictive of ALS progression [39], despite the existing conflicts between the different research studies carried out to date. Several investigations conducted in German [31] and French cohorts [32] reached the conclusion that the total cholesterol levels (TC) and hyperlipidemia in the former were associated with a positive effect on survival; in the latter, this was shown with an increment in low-density lipoprotein cholesterol (LDL) levels and/or a low concentration of high-density lipoprotein cholesterol (HDL). Through meta-analysis, a recent study has tried to establish a relationship between serum lipid levels and survival in patients with ALS, but it did not obtain enlightening results [40].

Despite the large volume of results, there are no conclusive data, and there is great controversy about the role of the serum lipid profile in ALS development and progression. For this reason, it is of great interest to clarify the complex relationship between dyslipidemia found in ALS patients and the rate/severity axis of disease progression. Understanding this relationship could help with both the early diagnosis of ALS and its prognosis and progression. We have accurately defined the relationship between plasma lipid levels in pre-symptomatic ALS patients with a family background, and it could be helpful in predicting both the overall symptoms and the rate of disease progression. Moreover, in the early stages of sporadic ALS, plasma lipid levels could clarify the degree of severity and the rate of progression of the disease, which is of great interest due to the lack of existing biomarkers for diagnosis and prognosis.

Apart from the possible relationship between blood lipid levels and ALS progression, it would be interesting to delve into the study of other possible blood molecules that set the foundation for disease development. Due to the convenience of blood collection, it represents the ideal method for the analysis of biomarkers as molecular indicators of dis-ease [41]. Moreover, in neurodegenerative diseases, the hemato-encephalic barrier might be compromised, increasing the possibility that these metabolites could be detected in the plasma.

Overall, the present study is focused on evaluating, with a meta-analysis methodology and systematic review, the results of previous investigations. First, we perform a systematic review with the aim of determining the differences in serum lipid levels between a control cohort and ALS patients. Then, we determine whether there is a relationship between serum lipid levels and disease progression. To determine the association between dyslipidemia in ALS patients and disease prognosis, we use a functional rating scale (ALS-FRS).

## 2. Materials and Methods

### 2.1. Design and Search Strategy

The review and meta-analysis were reported according to the reporting items for systematic reviews and meta-analyses (PRISMA) guidelines [42]. An exhaustive bibliographic search was carried out using several electronic databases: Dialnet, Scopus (Elsevier), and Medline (OVID). The search was performed in July 2022. The analyzed equation was (“triglycerides AND amyotrophic lateral sclerosis”, “triglycerides AND ALS”, “triglycerides AND motoneuron degenerative disease”, “serum lipid levels AND ALS”) or the keywords “ALS AND lipid levels” or “motoneuron disease AND lipid levels”, “lipid levels AND ALS-FRS” or “triglycerides OR cholesterol AND ALS-FRS”. We also performed a search using the Science Citation Index and Scopus to identify reports with citations of the identified articles, and a backward search was performed with the aim of revising and retrieving the references of selected studies for the systematic review and meta-analyses.

### 2.2. Eligibility Criteria

Studies were included according to the following criteria: (1) clinical studies, clinical trials, and randomized controlled trials; (2) a sample comprising ALS patients diagnosed by electromyography, muscle mass level, and specific symptomatology (dyspnea, dysarthria, loss of muscle strength, and loss of mobility) and a healthy population; (3) use of functional rating scale (ALS-FRS); (4) measurement of blood lipid molecules of disease diagnosis onset [1], total cholesterol (TC) in mg dL^−1^ or mmol L^−1^; (5) measurement of body mass index (BMI), symptom onset, age, and survival period given in months; and (6) full-text access. All studies that provided statistical data were included. The raw data units were passed in their entirety to mmol L^−1^. There was no restriction by publication date or language.

The exclusion criteria were (1) studies including other neurodegenerative pathologies or mixed data; (2) case reports; and (3) no inclusion of lipid levels. We did not place any restriction on age, gender, or co-morbidities in order to increase the number of results. The type of pharmacological treatment (analgesics or anti-inflammatories) in ALS patients at the time of diagnosis was not taken into account and was not one of the exclusion criteria. The sporadic or familial origin of the disease was not defined in any of the cases; therefore, it was not taken into consideration.

### 2.3. Selection Process: Study Variables and Data Extraction

Two independent reviewers analyzed the titles and abstracts and then the full texts according to the inclusion criteria. A third author was consulted in the case of discrepancies.

All the data were extracted and entered into an Excel table by two researchers independently; a third researcher was consulted when there was disagreement. All the discrepancies that emerged were resolved by discussion with the rest of the authors, with reference to the original study.

The following data were extracted from each study: (1) author, year, and location; (2) sample and mean age; (3) symptom onset; (4) BMI; and (5) serum lipid levels (Table 1). Corresponding authors were contacted for additional information when necessary. Two authors independently assessed the risk of bias in each included study according to the Cochrane Handbook for Systematic Reviews Interventions [43]. Specific domains were evaluated: selection bias (random sequence generation and allocation concealment), performance bias, incomplete outcome data, and “other issues”. We assessed each study as having “High risk of bias” or “Low risks of bias” in each domain.

### 2.4. Data Analyses

For the systematic review and the meta-analytic study, a data table was generated (see Table 1). The raw data units were passed in their entirety to mmol L^−1^. For the unit conversion, we used “Cholesterol Units Converter” based on research by Nordestgaard et al. [44]. Mean error and standard error were determined using extracted data from selected studies. For the meta-analysis, all the studies with statistical data were included. To determine the validity of the results, we only used those studies with a Spearman’s rho correlation. The effect size calculated was the meta-analytic estimate of the correlation between the TC, TG, LDL, and HDL levels and the ALS-FRS.

To assess the presence of heterogeneity in the studies, the I^2^ statistic was used; this is a value that quantifies the percentage of variability attributed to differences between studies. We used Egger’s test to assess the potential publication bias [45]. Basic statistical analyses (Student’s *t*-distribution and *t*-test) were performed using Systat Sigmaplot 11.0 Software, Inc. (Chicago, IL, USA). To carry out the meta-analysis, we used the Spearman correlation; specifically, we used the r value. StatsDirect Statistical Analysis Software v3.0 was used to perform the meta-analysis. A sensitivity analysis was performed to assess whether any study significantly varied the result of the meta-analysis.

**Table 1 ijms-24-08675-t001:** Characteristics of studies included in the systematic review. N/D: not available value.

				Cohorte	
Author	Year	Location	Article Type	N Control	N ALS
Ingre et al. [46]	2020	Sweden	Clinical trial	N/D	99
Mariosa et al. [47]	2017	Sweden	Prospective cohort study	N/D	623
Bjornevik et al. [23]	2021	USA	Randomized controlled trials	275	547
Chelstowka et al. [48]	2021	Poland	Clinical studies	N/D	203
Dorst et al. [31]	2011	Germany	Clinical trial	N/D	488
Ikeda et al. [33]	2012	Japan	Clinical trial	92	92
Won Yang et al. [49]	2013	Korea	Clinical trial	99	54.14
Mandrioli et al. [50]	2017	Italy	Clinical studies: retrospective cohort study	N/D	275
Dupuis et al. [32]	2008	France	Randomized controlled trials: retrospective cohort study	286	369
Huang et al. [51]	2014	China	Clinical studies	400	413
Ahmed et al. [45]	2018	Australia	Clinical studies	32	37
Dedic et al. [30]	2013	Serbia	Randomized controlled trials: retrospective cohort study	N/D	82
Nakamura et al. [52]	2022	Japan	Clinical studies: retrospective cohort study	N/D	78
Thompson et al. [53]	2021	UK	Longitudinal clinical studies: prospective population cohort	502,409	343
Chio et al. [29]	2009	Italy	Clinical studies	658	658
Nakatsuji et al. [54]	2017	Japan	Clinical trial	483	55
Ballantyne et al. [55]	1989	USA	Prospective, randomized clinical trial	N/D	39
Wuolikainen et al. [56]	2014	USA	Clinical trial	40	52
Sutedja et al. [57]	2015	The Netherlands	Randomized controlled trials	2100	303

## 3. Results

Our search yielded 61 documents, and 41 articles were excluded after reading the titles and abstracts because they had no connection to the study topic, or they were duplicated documents. Finally, 20 manuscripts remained and were included in the systematic review. Then, 16 were excluded from the meta-analysis because they did not include the inclusion criteria (Spearman’s rho correlation between ALS-FRS and the lipid levels). Finally, four studies were included in this meta-analysis. The selection process is detailed in Figure 1.

### 3.1. Included Studies and Lipid Levels in Control Cohort and ALS Patients

The basic data collection from the systematic review is shown in Table 1. As previously mentioned, 19 studies were selected in total. The selected articles were published between 1989 and 2022 (Table 1 and Table 2). There was demographic variety in the studies selected (European, American, and Asiatic). In some studies, the control population was not used (D/N). The type of article and the study are detailed in Table 1 and Table 2. The average ages of the control population and the ALS patients at the beginning of the disease did not show differences (58.7 and 59.6 years old, respectively). Most patients had non-bulbar (or spinal) symptom onset (71.1%), in comparison with the percentage of patients with bulbar symptom onset (28.9%). BMI was compared in order to more reliably compare the blood lipid levels between the control population and the ALS patients. Thus, the possible relationship between elevated serum lipid levels and the degree of obesity was eliminated. The average BMI was within the normal range and did not show a difference between the two populations (24.8 in the healthy population and 24.5 in the ALS patients) (Table 2).

In addition, the serum lipid levels were analyzed in the healthy population and the ALS patients in each of the studies selected (Table 3). The TC, LDL, HDL, and TG levels (mmol L^−1^) were extracted and converted from each of the manuscripts. Only studies showing lipid levels in mg·dL^−1^ or mmol L^−1^ were used in the comparison between both populations. So, 1 article was excluded in order to compare the serum lipid levels between the healthy population and ALS patients due to the lack of data in the manuscript. The data shown in Table 3 were used to perform the mean and standard error calculations in each population. No significant differences were obtained in any of the lipid levels (TC: *p* = 0.760; LDL: *p* = 0.598; HDL: *p* = 0.792; TG: *p* = 0.654) between the control cohort and the ALS patients. Nevertheless, an upward trend was observed in the ALS lipid levels (TC: 5.35 ± 0.2 mmol L^−1^; LDL: 3.15 ± 0.1 mmol L^−1^; HDL: 1.38 ± 0.1 mmol L^−1^ and TG: 2.45 ± 0.2 mmol L^−1^) compared with those of the healthy population (TC: 5.18 ± 0.3 mmol L^−1^; LDL: 3.08 ± 0.2 mmol L^−1^; HDL: 1.41 ± 0.1 mmol L^−1^ and TG: 2.33 ± 0.3 mmol L^−1^). These results could support the hyperlipidemia serum in ALS disease described by other authors. The mean survival data were extracted from each article and are shown in Table 3. The mean survival value was 31.28 months (~2.6 years) from the diagnosis of the disease. This value is relatively low because diagnosis of the disease is performed when advanced symptoms appear. The results presented in the table show that there is no relationship between increased lipid levels and a high survival rate.

**Figure 1 ijms-24-08675-f001:**
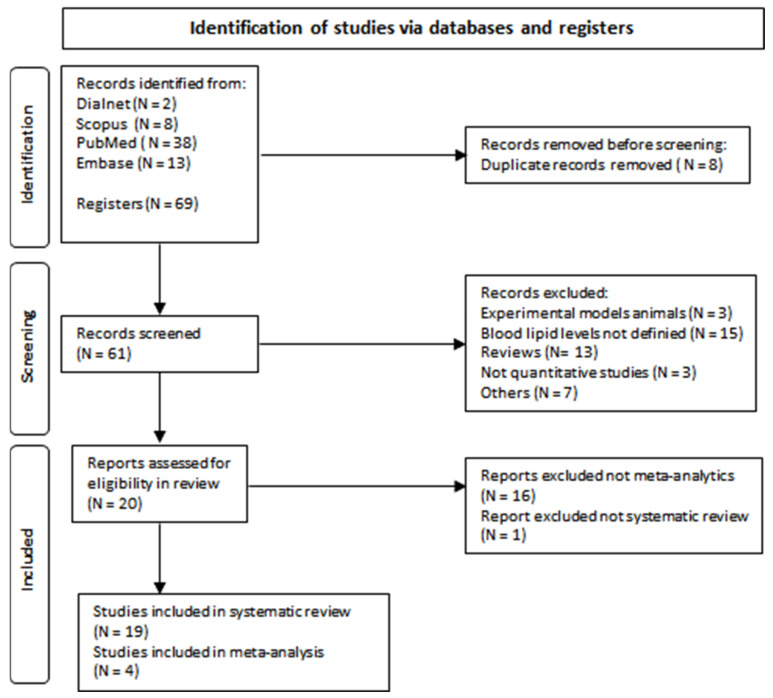
Flow diagram of document selection process.

### 3.2. Characteristics of Studies Included in Meta-Analysis

Only four studies were included for meta-analysis because they showed correlating data between the ALS-FRS level and the blood lipid levels. The data collected from the manuscripts used for meta-analysis had different origins: two studies were conducted in Japan, one in China, and one in Australia. The years of publication were between 2012 and 2018. In total, the entire sample comprised 597 ALS patients. According to the size of the cohort and the country of publication, 24.6% of the patients were Japanese, 69.2% were Chinese, and the remaining 6.19% were Australian.

### 3.3. Clinical Data

The ALS-FR score values were used to determine the initial disease progression. The ALS-FRS mean score was 36.6 ± 2.8 at the beginning of the disease. Life span was 37.8 ± 7.9 months (~3.15 years) after ALS was diagnosed. To strengthen our meta-analysis study and exclude the possibility that the differences obtained were due to changes in baseline lipid levels, we analyzed whether there was a significant difference in the TC and TG values between the control and the ALS subjects. The mean TC (Control: 208.7 ± 1.7 mg·dL^−1^ and 206.2 ± 1.5 mg·dL^−1^; *p* = 0.566) and TG (Control: 120.3 ± 4.2 mg·dL^−1^ and 147.6 ± 7.4 mg·dL^−1^; *p* = 0.159) levels were not statistically significant in the ALS patients compared to those of the control group. These data are represented in Table 4 and Table 5. However, an interesting increase in TG levels, but not in TC levels, was observed in the ALS patients when compared with those of the control population.

### 3.4. Meta-Analysis for ALS-FRS Score and Lipid Levels

Four random-effect meta-analyses were performed. The basic data are shown in Table 4. Each meta-analysis analyzed the correlation between the TG, LDL, HDL, and TC levels and ALS progression (ALS-FRS score). The data are shown in Table 5. Three studies were used to correlate the TC and TG levels with the ALS-FRS scores. Moreover, two of them were used to perform the meta-analysis of the HDL and LDL levels with the ALS-FRS score.

The estimated meta-analytical correlation of the ALS-FRS value with TG was r = −0.13 (95% CI −0.35, 0.10. *p* > 0.05; *n* = 505) (Figure 2); with LDL, it was r = −0.26 (95% CI 0.64, 0.23. *p* > 0.05; *n* = 129) (Figure 3); with HDL, it was r = 0.17 (95% CI −0.19, 0.49. *p* < 0.05; *n* = 468) (Figure 4); and, finally, with TC, it was r = −0.20 (95% CI −0.59, 0.27. *p* > 0.05) (*n* = 560 ALS patients) (Figure 5). I^2^ was higher than 50%, reflecting a high degree of heterogeneity in all the meta-analyses.

No study was eliminated after the sensitivity analysis, and the value of the Egger test showed that there was no publication bias. As one of the main parameters used in the meta-analysis was the ALS-FRS scale, the size of the control cohort (healthy individuals) was not considered for our meta-analysis study.

## 4. Discussion

Lipid metabolism is essential in the central nervous system for its correct functioning. Thus, alterations in lipid metabolism play major roles in neurological disease. Apart from ALS, these alterations have been demonstrated in other pathologies: spinal muscular atrophy [58], spinocerebellar ataxia [59], Huntington’s disease [60], Parkinson’s disease [61], and Alzheimer’s disease [62,63]. In the latter two pathologies, a strong inverse relationship between survival and basal lipid levels has been observed, in contrast to the linear relationship between basal plasma lipids and ALS described in the literature [17,43,64,65,66].

Interestingly, most of the lipid metabolism alterations appear at the ages of 50–55 years, which coincides with the average age of ALS onset. These lipid alterations observed in ALS patients could lead to neuronal degeneration, although little is known about the correlation between dyslipidemia and disease severity. In this paper, we analyzed the relationship between disease development and the alterations in plasma lipid levels (total cholesterol, triglycerides, HDL, and LDL). The ALS-FRS scores from all the studies were used to define the rate of progression in early-stage disease and to predict survival [47]. The results of this systematic review showed high serum lipid levels in ALS patients in comparison with those of the control cohort in most of the articles included in this paper. These differences were more pronounced in the cholesterol, TG, and LDL levels, while the HDL levels did not present differences between the two cohorts. However, we found no relationship between lifespan (mean survival) and high serum lipid levels (HDL, TC, and TG). In contrast to these results, Dorst et al., determined that ALS patients with elevated TG and TC serum levels had a prolonged survival. Nevertheless, this work did not provide the raw data to compare and discuss its results [31]. In this context, Dupuis et al. demonstrated an abnormally elevated LDL/HDL ratio, which increased survival by more than 12 months [32]. Moreover, Thompson et al. observed that high blood HDL was related to a reduction in the risk of developing ALS [53]. In contrast, Nakamura et al. showed that higher HDL levels were associated with poor prognosis in ALS patients. This study had the limitation of being a single-center retrospective study with only 78 ALS patients. On the other hand, the meta-analysis data showed that there was no significant difference in the TC, TG, HDL, and LDL levels in any of the ALS patients at the beginning of the disease compared to those of the control subjects. This corresponds with the data published by other authors, such as Bjornevik et al., Bouteloup, and Dorst et al. [23,28,51]. However, research by other authors argues for a relationship between baseline plasma lipids and the degree of disease development [54,67]. In this regard, it has been observed that plasma lipid levels are altered in the early phase of ALS, but once the disease is established, the levels normalize, which can be used for early diagnosis [57]. This is the reason why the baseline lipid concentrations in our study were taken after the disease onset. It would have been useful to clarify the type of relationship between the TC, TG, HDL, and LDL levels and the rate of disease progression in order to establish this relationship as a methodology in the assessment of the prognosis and severity of ALS. However, this information is currently lacking in the literature, and further studies should be conducted to assist in the management of this fatal disease.

Several controversies have been described in relation to lipid levels and the status of ALS patients. Indeed, some studies have revealed that ALS patients with low plasma lipid levels show a higher survival rate [32,68]. These studies found that hyperlipidemia has a neuroprotective effect in subjects prone to the development of the disease [29,31,33,69,70,71,72]. Recently, a new meta-analytic study was published showing that ALS patients from Europe and Asia had lower levels of TG and HDL and that ALS patients from Europe had higher levels of TC and LDL, but only the TC levels in Asian ALS patients were significant. However, the overall survival of ALS patients was not correlated with lipid levels [73]. In any case, it seems that plasma lipid levels, the type of dietary fatty acids ingested, environmental factors, and exercise may randomly affect the development of ALS [33].

The discovery of new markers that would bring us closer to a more precise clinical treatment or to the development of new drugs would greatly support the fight against the disease and would increase patients’ life expectancy, as well as offer a considerable improvement in their quality of life [41]. Numerous physiological alterations have been observed in ALS development in the early stages: protein aggregates in cytoplasmic motoneurons, glutamate, increased levels of neurofilament light chain and phosphorylated neurofilament heavy chain in the cerebrospinal fluid, or decreased neuron growth factor in muscles. All alterations need invasive techniques to be determined (neurobiopsy, spinal fluid extraction by lumbar puncture, or muscle biopsy). The use of markers that are easily measured using primary care techniques is a priority in mortal disease. In this sense, changes in lipid metabolism have been observed in both human and animal ALS models [46,74]. In addition, new investigations have focused their objective on the design of drugs capable of modulating the enzymatic activity of lipid metabolism [75]. In this regard, higher resting energy expenditure was observed in a large population of ALS patients at rest (30–60% present hypermetabolism) [76,77]. In addition, a large percentage of ALS patients present glucose intolerance, increased muscle glucose uptake, high oxygen consumption, and high lactate output [78]. Regarding BMI, we did not observe differences in this parameter in either of the cohorts; the results showed similar values (index ~24), suggesting that hyperlipidemia in ALS patients is not related to lifestyle or diet. Similar results were obtained in a clinical trial in Korea, which found no association between BMI and disease severity using ALS-FRS. Furthermore, they concluded that nutrient intake decreases with disease progression in most ALS patients [79]. Supporting the obtained results, a Korean clinical trial evidenced no association between BMI and disease severity using ALS-FRS. In addition, they concluded that the intake of nutrients decreased with disease progression in most ALS patients. In this context, several diets have been proposed to alleviate the symptoms of the disease, but the results are still not very clear. The intake of antioxidants derived from vegetables and legumes has a beneficial effect on delaying disease progression [80]. Other research has evidenced that complex vitamins, a ketogenic diet, or a high-fat diet could be beneficial in ALS treatment [81,82]. However, contrary to our results, some authors have found a relationship between BMI and ALS-FRS, suggesting that baseline BMI may help to predict disease progression in ALS patients (higher BMI was associated with slower ALS-FRS). According to these authors, faster deterioration and disease progression were predicted in patients with a BMI higher than 30 [39].

All the investigations included in our meta-analysis study were carried out in eastern areas (Japan, China, and Australia), which highlights an important limitation due to the great differences in lifestyles. This includes a powerful demographic bias that can influence the data interpretation. Another limitation was the low number of studies available, as well as the number of ALS patients under study. Due to these issues, the data obtained with our study should be interpreted with caution. Nevertheless, our study does not disregard a potential relationship between basal lipid levels in plasma and ALS onset, which will be of the utmost importance in the pre-diagnosis of the disease. However, we were not able to obtain a clear conclusion regarding the relationship between basal lipid levels and the rate of ALS progression once the symptoms had already been established. Several evaluations are required to determine ALS pre-diagnosis in the pre-symptomatic stages. Although an ALS gene mutation is present in patients, evidence for the absence of motor neuron dysfunction is required. Moreover, neuromuscular examinations by ALS experts and electromyography are needed to help in the pre-diagnosis of the disease [22,83]. The absence of specific biomarkers necessitates relevant research on the pre-diagnosis of the disease.

Another of the main limitations we faced in the development of this work was the fact that most of the research was performed using transgenic ALS animal models; this excluded the mechanisms and causes of most ALS types (familial ALS) because research involving humans is less frequent and more complex due to the difficulty in detecting the disease at an early stage of development. Unfortunately, the majority of ALS patients are diagnosed late when motor dysfunction symptoms appear. Most of the pharmacological treatments used in ALS are symptomatic treatments: spasticity (muscle relaxants or cannabinoids [84]), sialorrhoea (hypersalivation) (anticholinergic drugs), pain (tricyclic antidepressants, non-steroidal anti-inflammatory drugs, opioids, and cannabis), respiratory insufficiency (non-invasive ventilation), and others [5]. In addition, great interest has been placed in research on drugs that act on ion channels and on skeletal muscle [85,86]. These authors have even established the possible relationship between ALS, muscle atrophy, and the taking of statins (atorvastatin) (drug used to lower serum cholesterol levels) [87].

## 5. Conclusions

In conclusion, we were able to confirm that dyslipidemia exists in ALS patients at the disease onset, with hyperlipidemia being the most common alteration of lipid levels in these patients. However, the cause of hyperlipidemia is unknown: on the one hand, high lipid levels may be a consequence of motor neuron degeneration, and on the other hand, high lipid levels could be a neuroprotective mechanism that is activated in response to motor neuron degeneration. In both cases, the use of serum lipid levels as biomarkers has great clinical potential. Better monitoring of lipid level parameters, taking into account the close correlation with the ALS-FRS score, is needed throughout the course of the disease until the patient’s death. On the other hand, the results obtained in the meta-analysis led us to consider the need for more research on the onset of ALS to increase the number of studies used in any future meta-analysis, as well as the number of ALS patients involved. Therefore, the discovery of new markers, such as lipid alterations, or the development of new drugs will greatly support the fight against the disease and will contribute to an increase in the life expectancy of patients, as well as offer a considerable improvement in their quality of life.

## Figures and Tables

**Figure 2 ijms-24-08675-f002:**
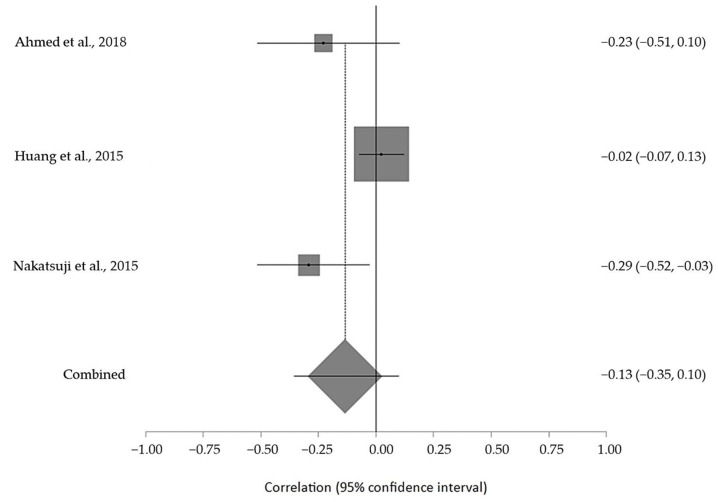
Forest plot of serum TG in ALS patients and ALS-FRS score [45,51,54].

**Figure 3 ijms-24-08675-f003:**
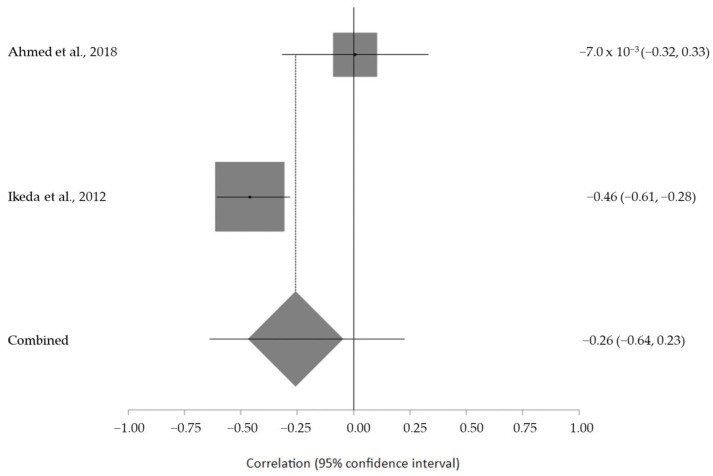
Forest plot of serum LDL in ALS patients and ALS-FRS score [33,45].

**Figure 4 ijms-24-08675-f004:**
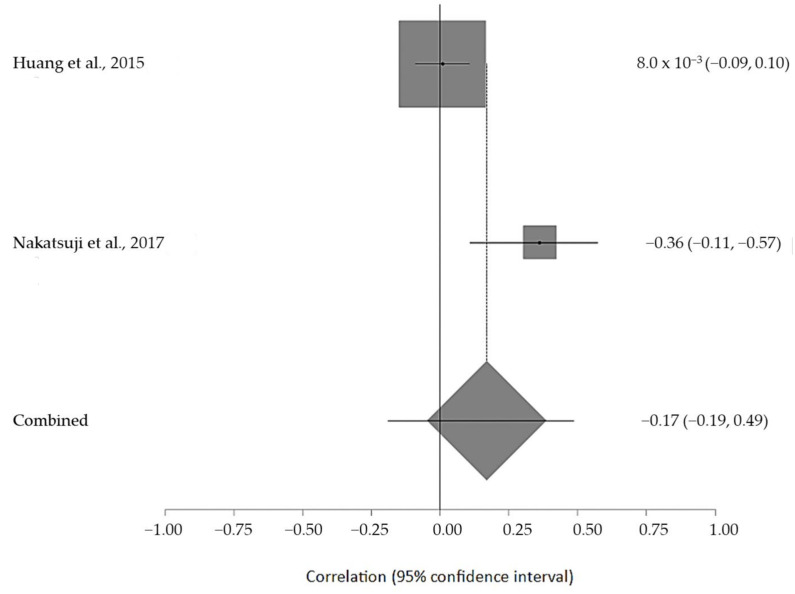
Forest plot of serum HDL in ALS patients and ALS-FRS score [51,54].

**Figure 5 ijms-24-08675-f005:**
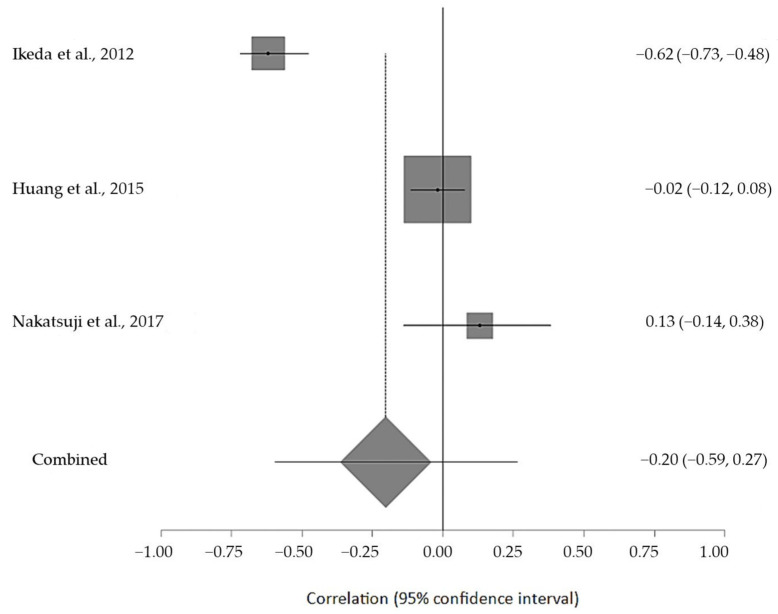
Forest plot of serum TC in ALS patients and ALS-FRS score [33,51,54].

**Table 2 ijms-24-08675-t002:** Basic data of studies included in the systematic review. N/D: not available value.

					Site of Symptom Onset		BMI	
Author	Year	Location	Age Control	Age ALS	Bulbar	Nonbulbar	Control	ALS
Ingree et al. [46]	2020	Sweden	N/D	65.7	38	61	N/D	24.38
Mariosa et al. [47]	2017	Sweden	N/D	67	N/D	N/D	N/D	N/D
Bjornevik et al. [23]	2021	USA	64.6	69.4	N/D	N/D	26.9	26.2
Chelstowka et al. [48]	2021	Poland	N/D	56	N/D	N/D	N/D	24.6
Dorst et al. [31]	2011	Germany	N/D	57.6	89	398	N/D	25.4
Ikeda et al. [33]	2012	Japan	59.2	58.8	10	82	22.8	22.6
Won Yang et al. [49]	2013	Korea	52.5	54.1	N/D	N/D	N/D	N/D
Mandrioli et al. [50]	2017	Italy	N/D	65.2	83	30.2	N/D	24.5
Dupuis et al. [32]	2008	France	N/D	57.5	92.2	276.7	N/D	24.6
Huang et al. [51]	2014	China	51.4	51.8	N/D	N/D	21.5	21
Ahmed et al. [45]	2018	Australia	64.7	55.9	9	28	24.9	25.7
Dedic et al. [30]	2013	Serbia	N/D	53.7	30	52	N/D	26.7
Nakamura et al. [52]	2022	Japan	N/D	71	26	52	N/D	21.7
Thompson et al. [53]	2021	UK	58	62	N/D	N/D	26.7	27.2
Chio et al. [29]	2009	Italy	62.7	62.9	201	457	24.8	25.1
Nakatsuji et al. [54]	2017	Japan	53.2	51.1	N/D	N/D	24.2	22.7
Ballantyne et al. [55]	1989	USA	N/D	50	N/D	N/D	N/D	N/D
Wuolikainen et al. [56]	2014	USA	61.7	58.7	N/D	N/D	25.3	23.8
Sutedja et al. [57]	2015	The Netherlands	59	64	90	205	26	25
			58.7 ± 1.5	59.6 ± 1.3	28.9%	71.1%	24.8 ± 0.6	24.5 ± 0.5

**Table 3 ijms-24-08675-t003:** Serum lipid (TC, LDL, HDL, and TG in mg·dL^−1^) levels in control population and ALS patients. Mean values are shown in the last row; the values represent the mean and standard error. The Student’s *t*-test was performed. N/D: not available value.

	Total Cholesterol (mmol·L^−1^)		Low-Density Protein (mmol·L^−1^)		High-Density Protein (mmol·L^−1^)		Triglycerides (mmol·L^−1^)		
Author	TC Control	TC ALS Patients	LDL Control	LDL ALS Patients	HDL Control	HDL ALS Patients	TG Control	TG ALS Patients	Mean Survival (Months)
Ingre et al. [46]	N/D	5.46	N/D	3.14	N/D	1.64	N/D	1.54	13.72
Mariosa et al. [47]	N/D	5.48	3.59	3.69	N/D	1.52	N/D	N/D	12.00
Chelstowka et al. [48]	N/D	5.37	N/D	3.28	N/D	1.34	N/D	3.50	19.92
Dorst et al. [31]	4.70	6.00	4.91	3.87	N/D	1.29	1.40	1.77	51.00
Ikeda et al. [33]	5.33	5.47	3.21	3.34	1.49	1.49	3.06	3.30	23.70
Won Yang et al. [49]	5.11	4.87	3.11	2.99	1.20	1.22	4.04	3.28	N/D
Mandrioli et al. [50]	N/D	5.12	N/D	3.36	N/D	1.29	N/D	2.59	N/D
Dupuis et al. [32]	2.10	2.50	1.20	1.60	0.60	0.60	1.30	1.30	N/D
Huang et al. [51]	5.31	5.24	2.81	2.80	1.36	1.20	3.14	3.30	21.80
Ahmed et al. [45]	5.51	6.60	N/D	N/D	1.90	1.50	1.00	1.90	20.40
Dedic et al. [30]	N/D	5.80	N/D	2.95	N/D	1.37	N/D	1.87	50.52
Nakamura et al. [52]	N/D	N/D	N/D	2.97	N/D	1.63	N/D	2.82	N/D
Thompson et al. [53]	5.65	5.64	3.52	3.54	1.40	1.30	1.48	1.67	14.63
Chio et al. [29]	5.38	5.46	3.25	3.33	1.54	1.53	3.05	2.98	N/D
Nakatsuji et al. [54]	5.56	5.30	N/D	N/D	1.45	1.54	3.66	3.76	85.20
Ballantyne et al. [55]	N/D	5.26	N/D	3.05	N/D	1.02	N/D	2.65	N/D
Wuolikainen et al. [56]	5.80	6.00	3.20	3.40	1.75	1.85	1.25	1.10	N/D
Sutedja et al. [57]	5.85	5.50	3.90	3.20	1.45	1.55	N/D	N/D	N/D
	5.18 ± 0.3	5.35 ± 0.2	3.08 ± 0.2	3.15 ± 0.1	1.41 ± 0.1	1.38 ± 0.1	2.33 ± 0.3	2.45 ± 0.2	31.28 ± 7.46

**Table 4 ijms-24-08675-t004:** Basic data of studies included in meta-analysis. * Last row values represent the mean and standard error and total bulbar and non-bulbar symptoms onset. N/D: not available value.

Author	Location	Participants (ALS/Control)	Age (Years) (ALS/Control)	Symptom Onset (Bulbar/Nonbulbar)	BMI (ALS/Control)
Ikeda et al., 2012 [33]	Japan	92/92	58.8/59.2	10/82	22.6/22.8
Huang et al., 2015 [51]	China	413/400	51.8/51.4	N/D	21/21.5
Ahmed et al., 2018 [45]	Australia	37/32	55.9/64.7	9/28	25.7/24.9
Nakatsuji et al., 2017 [54]	Japan	55/483	51.1/53.2	N/D	22.7/24.2
			* 54.4 ± 0.9/57.1 ± 1.5	19/110	

**Table 5 ijms-24-08675-t005:** Characteristics of studies included in meta-analysis. Last row values represent the mean and standard error and total bulbar and non-bulbar symptoms onset. N/D: not available value.

	Total Cholesterol (mmol·L^−1^)		Triglycerides (mmol·L^−1^)		
Author	TC Control	TC ALS	TG Control	TG ALS	ALS-FRS
Ikeda et al., 2012 [33]	5.33	5.47	3.06	3.30	40.3
Huang et al., 2015 [51]	5.31	5.24	3.14	3.30	31.2
Ahmed et al., 2018 [45]	5.51	6.60	1.00	1.90	38.5
Nakatsuji et al., 2017 [54]	5.56	5.3	3.66	3.76	N/D
	5.42 ± 0.1	5.65 ± 0.3	2.71 ± 0.5	3.06 ± 0.4	36.6 ± 2.8

## Data Availability

Not applicable.

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
