# Peer review of "Amyotrophic Lateral Sclerosis and Serum Lipid Level Association: A Systematic Review and Meta-Analytic Study"

_ijms, 2023, doi:10.3390/ijms24108675_

Round 1
Reviewer 1 Report
This metanalysis is aimed at identifying potential relationships between lipids and ALS and is therefore a useful piece of research to identify if there is any robust relationship. There is a fairly comprehensive discussion of the literature. There are some methodological issues that need addressing in the Manuscript. Please also check carefully the calculations and numbers as there appear to be some errors.
1. As the authors likely are aware, an extremely similar review was published pretty recently, the authors should cite it, understandably the findings are identical given the similar methods; https://n.neurology.org/content/100/10/e1062
2. I am not sure I understand the rationale for excluding studies that don't give the cholesterol in mg/dl why not just convert the cholesterol from mmol/L if given.
3. Table 4 has an issue, the value for the Ahmed study seems unlikely and I don't think its correct, that paper is also in mmol/L? Please correct this and corresponding numbers, correct discussion etc.
4. Figure 5 does not exist?
Minor:
Figure 1 needs a careful spellcheck
Author Response
Response: We would like to thank you for your helpful comments on the manuscript. We are sincerely grateful to you for the time spent reading, commenting, and suggesting changes and modifications, which have greatly improved the quality of the work. We believe that the revisions made based on your reports have significantly improved the original manuscript.
Major revision
This metanalysis is aimed at identifying potential relationships between lipids and ALS and is therefore a useful piece of research to identify if there is any robust relationship. There is a fairly comprehensive discussion of the literature. There are some methodological issues that need addressing in the Manuscript. Please also check carefully the calculations and numbers as there appear to be some errors.
- As the authors likely are aware, an extremely similar review was published pretty recently, the authors should cite it, understandably the findings are identical given the similar methods; https://n.neurology.org/content/100/10/e1062
Response: Thanks for the recommendation. Recently published study has been cited both in the introduction and in the discussion (Reference 38). This new publication does not establish conclusive results, but it does help to argue the possible relationship between the lipid profile of ALS patients and the progression degree of the disease. We thank you very much for indicating it to us.
- I am not sure I understand the rationale for excluding studies that don't give the cholesterol in mg/dl why not just convert the cholesterol from mmol/L if given.
Response: Thanks for the comment. We decided to use only the raw data extracted from the articles in mg·dL-1 and not use unit conversions. Thus we have worked with the real published data without manipulating it. When we tried to do some units conversion, the values obtained were very different from those compared in the rest of the manuscripts. For this reason, we use the actual raw data that the authors provide us with in their publications. This adds a major inconvenience by greatly decreasing the number of values and scripts used. In addition, the units most commonly used in the clinic for serum lipid measurements are mg·dL-1.
- Table 4 has an issue, the value for the Ahmed study seems unlikely and I don't think its correct, that paper is also in mmol/L? Please correct this and corresponding numbers, correct discussion etc.
Response: Thanks for the comment. Data from Ahmed et al. were removed to calculate total cholesterol in the control cohort and the diseased cohort. Mean and error values already appear in the text excluding Ahmed's data (208.7 ± 1.7 and 206.2 ± 1.5 mg·dL-1, respectively). It was a mistake not to remove this data from Ahmed et al. in the table 4.
Although in the original article by Ahmed et al. indicates that the units are mg·dL-1, it was decided to eliminate them because these data were very different compared to those published in other publications. We thought it was more reliable to remove the total cholesterol data from the averages shown in Table 4.
- Figure 5 does not exist?
Response: Thanks for the comment. Indeed, an editing error was made and figure 5 was not included. It has been included after figure 4. We are very sorry for the editing error.
Minor revision
Figure 1 needs a careful spellcheck
Response: Thanks for the comment. The spelling errors in Figure 1 have been corrected.
Reviewer 2 Report
The review of Teresa Pardo-Moreno et al. is an interesting study that classify the bibliography in which plasma lipids level are described as biomarker for ALS diagnosis. However the examined papers are few and often contradictory. The manuscript need to be better described, in particular the discussion is a collection of data that should be better organized.
Moreover, there are many spelling mistakes to correct. Here are some of them:
Line 70 iduced
Line 73 ex-citatory
Line 99 of-ten
Line 196 correct
Line 213 correct tittles
Line 224 correct articles
And other errors to check.
Major points:
Line 79-80 this concept has been already explicited
Line 101 here, it seems that the great part of the authors concord with the finding that ALS is associated with hypolipidemia and decline in BMI
Line 102 I think this is not an opposite conclusion
The Discussion need to be importantly revised
Line 337 ethology incorrect
Line 341 which model?
Line 362 a large…….population (please, add)…… of ALS patient
Line 366 the same results emerged….. (please change obtained with emerged, these are not your results)
Line 370 correct in: life span
Line 371 Dorst et al….. this phrase is difficult to understand 372 “we do not provide the raw……” is correct?
There are other reviews describing that it is still unclear the role of lipids increase. However, it is necessary to pay attention to the use of lipid lowering drugs, particularly in the earlier phase of the disease. The authors can cite these ms: doi: 10.3390/cells11030415, PMID: 35159225; doi: 10.1016/j.taap.2016.06.032, PMID: 27377005; doi: 10.1111/bph.15276, PMID: 32986860
Author Response
We would like to thank you for your helpful comments on the manuscript. We are sincerely grateful to you for the time spent reading, commenting, and suggesting changes and modifications, which have greatly improved the quality of the work. We believe that the revisions made based on your reports have significantly improved the original manuscript.
Minor points
The review of Teresa Pardo-Moreno et al. is an interesting study that classify the bibliography in which plasma lipids level are described as biomarker for ALS diagnosis. However the examined papers are few and often contradictory. The manuscript need to be better described, in particular the discussion is a collection of data that should be better organized.
Moreover, there are many spelling mistakes to correct. Here are some of them:
Line 70 iduced
Line 73 ex-citatory
Line 99 of-ten
Line 196 correct
Line 213 correct tittles
Line 224 correct articles
And other errors to check
We greatly appreciate your review and suggestions to improve the manuscript. We are very sorry for the editing error. Phrasing of manuscript was reviewed in detail and modified to improve its reading and comprehension. Proposed spelling changes have been modified in its whole.
Major points
Response: thanks for the comments. All comments have been considered and modified.
Line 79-80 this concept has been already explicated.
Line 79-80: This concept has been pointed out to indicate that the use of transgenic or knock out animal models is not fully representative for studying ALS, because mort of cases are sporadic. With this explanation we want to give importance to the research carried out on ALS patients and patients without discrediting the experimentation carried out on genetically modified animal models.
Line 101 here, it seems that the great part of the authors concord with the finding that ALS is associated with hypolipidemia and decline in BMI
Response: Thanks for your comment. The phrase has been rewritten.
Line 102 I think this is not an opposite conclusion
Response: Thanks for your comment. The meaning of the sentence has been changed.
The Discussion need to be importantly revised.
Response: We greatly appreciate your review and suggestions to improve the discussion. We have reorganized the entire discussion and removed information that we felt was not of interest. The initial part of the discussion has been moved to the end of the discussion. In addition, new references of interest have been included. In this way we believe that the text is more understandable to average readers.
Line 337 ethology incorrect Corrected
Line 341 which model? OK
Line 362 a large…….population (please, add)…… of ALS patient OK
Line 366 the same results emerged….. (please change obtained with emerged, these are not your results) OK
Line 370 correct in: life span OK
Line 371 Dorst et al….. this phrase is difficult to understand 372 “we do not provide the raw……” is correct? Wording has been changed.
There are other reviews describing that it is still unclear the role of lipids increase. However, it is necessary to pay attention to the use of lipid lowering drugs, particularly in the earlier phase of the disease. The authors can cite these ms: doi: 10.3390/cells11030415, PMID: 35159225; doi: 10.1016/j.taap.2016.06.032, PMID: 27377005; doi: 10.1111/bph.15276, PMID: 32986860
Response: Thanks for your recommendation. We have included news articles proposed in discussion. News references have been included in the discussion like 76, 78 and 76 respectively.
Reviewer 3 Report
Identification of preclinical biomarkers of ALS is crucial and the study is important in that context.
The text contains lot of formal mistakes considering English language and it is strongly suggested to have the entire text reviewed by native English speaker.
Sentences in following lines need to be reformulated because they are hard to understand. (65, 76-79, 81-82, 83-86, 99, 111-115, 127-129, 134-135, 246-248, 273-276, 460-465).
Plus there are lot of typos, so entire text need to be reviewed.
Change the text:
Line 337 - 'ethology' is study of animal behavior.
Line 387 - 'Worth nothing'
Furthermore, tables should be mentioned in text chronologically. However Table 4 is mentioned before Table 3.
And Figure 5 is missing.
Author Response
Identification of preclinical biomarkers of ALS is crucial and the study is important in that context.
The text contains lot of formal mistakes considering English language and it is strongly suggested to have the entire text reviewed by native English speaker.
Response: We would like to thank you for your helpful comments on the manuscript. Thanks for the comments and suggestions. First, an extensive revision of the English language and style was carefully done. We believe that the revisions made based on their reports have significantly improved the manuscript. A English language editing by MDPI has modified the manuscript. The text has been checked for correct use of grammar and common technical terms.
Sentences in following lines need to be reformulated because they are hard to understand. (65, 76-79, 81-82, 83-86, 99, 111-115, 127-129, 134-135, 246-248, 273-276, 460-465).
Response: All proposed phrases and lines have been modified. Wording has been changed and we believe that the text is more understandable to average readers.
Change the text:
Line 337 - 'ethology' is study of animal behavior. OK
Line 387 - 'Worth nothing' OK
Furthermore, tables should be mentioned in text chronologically. However Table 4 is mentioned before Table 3.
Response: Thanks for the comment. The tables are now mentioned in the text chronologically. We are very sorry for the editing error.
And Figure 5 is missing.
Response: Thanks for the comment. Indeed, an editing error was made and figure 5 was not included. It has been included after figure 4. We are very sorry for the editing error.
Reviewer 4 Report
Unfortunately, I think that the manuscript in this form does not meet the required criteria for publication. I hope that my suggestions could improve the methodological formulation, in order to make your article more impactful.

Author Response
General concept comments The authors review in an incomplete manner the topic announced in the title, lacking of methodology strategy and the correct contextualization of the article. The need of linking neurodegeneration and lipid metabolism is growing and complex, but in literature there are already some similar works (please refer to PMID 36836867, 35625841, 31562633.. for recent examples).
Response: We would like to thank you for your helpful comments on the manuscript. We are sincerely grateful to you for the time spent reading, commenting, and suggesting changes and modifications. We believe that the chosen title includes the methodology used and indicates the main objective and hypothesis of the manuscript. News references have been included like reference 66 (PMID 36836867) and 39 (31562633). This reference 35625841 was already included like reference 72.
If the strength point is the comparison with the healthy population, could be useful to be more concise and focused on this argument. Thereafter, there are many linguistic errors that make the text difficult to be read and followed. Moreover, in introduction there are many errors in reporting the knowledge in ALS etiology and epidemiology, often without the correct cited references.
Response: Thanks for the comment. We have tried our best to carefully consider and respond all the questions raised. As we have said before, an exhaustive revision and modification of the language has been carried out by a person with knowledge of English. In this way we believe that the text is more understandable to average readers. The introduction have been revised and checked and large changes were performed.
The presentation of the methods are not linear, and also the results are confusing (ie: the “survival time” in 3.3 to what is referred? In summary, this manuscript lacks sufficient numbers of appropriate search in literature for lipid metabolism, in order to evaluate its utility in ALS. For example, a more proper search criteria should include also low-density lipoprotein (LDL), high-density lipoprotein (HDL), lipoprotein a, statin, phospholipids, fatty acid... The authors do not combine these key-words to evaluate the extended impact for lipid alterations on ALS risk; the statistical method is weak. As such, the discussion regarding the significance of lipid measures remains speculative and not supported by sufficient data and methods.
Response: Thanks for the comments. The methodology is presented in the same order as the results. “Survival time” was replaced by “life span”. We thought that including other terms such as LDL or HDL in the search criteria was redundant. The use of the criteria mentioned in the methodology already included the criteria to which you refer. Still, we thank you for your insight.
Specific major comments (abstract and introduction)
- Abstract: from the journal guidelines: The abstract should be a single paragraph and should follow the style of structured abstracts, but without headings. Please correct it.
Response: Thanks for the comment. Abstract was corrected and headings were removed.
- Abstract, line 22: “as a method of assessment” what the authors intend for this expression?
Response: Thanks for the comment. “as a method of assessment” was replaced by “pre-diagnostic”.
- Abstract, lines 25-26: the search strategy reported here is not correct in comparison with methods section
Response: We added “its variants” after “triglycerides AND amyotrophic lateral sclerosis”
- Abstract, line 32: “an expansion of the geographical area would be of interest.”: which geographic area could be more interesting?
Response: Thanks for question. We refer to other geographical areas than those used in the articles included in the meta-analytic study, which are all eastern areas.
- Introduction, lines 38-39: “The cause of ALS is unknown and is considered a heterogeneous syndrome”. Please cite the references. Moreover, what the author intend for a heterogeneous syndrome? From a clinical perspective? Genetical one? Or pathophysiologically?
Response: The referenced was included like 2. We refer a “heterogeneous syndrome” from perspective with different pathophysiological causes.
- Introduction, line 39 “Patients with ALS have a maximum life expectancy of 5 five years after diagnosis”: not properly true. The authors seem to contradict this concept afterwards along the text (lines 48-49).
Response: Thanks for the comment. To avoid misinterpretation we have modified "life expectancy of" by "life expectancy up"
- Introduction, lines 40-45: the data presented are not correct (Also true and contradictory for lines 79-80) Please verify the correct percentages for sporadic ALS, familial ALS and their relative genetic background. OK
- Introduction: lines 45-48: there are not any reference cited in support of the epidemiological data.
Response: A new reference was added after reference 4.
Introduction: lines 49-54: which is the purpose of presenting all this information to the lecturer? In my opinion are not focused on the article. Moreover, are not correctly reported in content and form.
Response: Thanks for the comment. This information helps the reader not related to ALS to understand the severe symptomatology of the disease.
- Introduction: lines 60-75: the knowledge on ALS pathomechanism is complex to report in its totality. In this case, the flow of speech is very confusing and not finalized to the objective of the studye.
Response: ALS pathomechanism helps the reader not related to ALS to understand the disease.
- introduction, lines 76-80: please verify the property of the information written and add the relative references.
Response: the reference was added.
- introduction, lines 81-82: this sentence is not easy to understand.
Response: wording has been changed.
-introduction, lines 82 and following: EEC criteria revised!! What reported is confusing in its formulation. OK
- introduction, lines 99-100: dyslipidemia as a milestone for ALS? In which declination?
Response: Thanks for the comment. There are controversies with this observation. It is only known that there are alterations in lipid metabolism in patients with ALS but not the direction. For this reason we use the term dyslipidemia.
- introduction, lines 103-104: calling “familial ALS” for animal models is not proper.
Response: Thanks for the comment. The term was removed.
- introduction, lines 105-108: not clear the relationship between lipids and ALS risk
Response: One of the goals of the manuscript is to establish the relationship and direction between lipid levels and ALS. Based on the bibliography we found results from both sides.
- introduction lines 113-115: not clear what the French study have found for lipid levels. OK
- introduction, lines 126-130: the author descript in two different sentences, divided by “on the other hand” the same concept. Please simplify and be more concise
Response: Thanks for the comment. Sentences were modified.
- introduction, lines 131 – 133: which references? Why is it important for plasmatic lipid? Are they product from the CNS?
Response: Indeed, neuronal degeneration can induce an increase in lipids level by degradation of the plasma membrane.
Below are some references to specific neuronal diseases show relation between high lipid level in central nervous system and degenerative progress:
- Alzheimer’s disease: DOI: 1074/jbc.274.52.37046; DOI: 10.1016/j.neulet.2006.02.008
- dementia associated with type 2 diabetes: DOI: 2174/1567205013666161201200722
- psychiatric disorders: DOI: 15252/emmm.201505749; DOI: 10.15252/emmm.201505677; DOI: 10.1007/s00702-014-1289-9
- Others:
TOKUMURA, A., KANAYA, Y., KITAHARA, M., MIYAKE, M., YOSHIOKA, Y. & FUKUZAWA, K. 2002. Increased formation of lysophosphatidic acids by lysophospholipase D in serum of hypercholesterolemic rabbits. J Lipid Res, 43, 307-15.
GUPTA, S., KNIGHT, A. G., LOSSO, B. Y., INGRAM, D. K., KELLER, J. N. & BRUCE-KELLER, A. J. 2012. Brain injury caused by HIV protease inhibitors: role of lipodystrophy and insulin resistance. Antiviral Res, 95, 19-29.
ZHAO, Z., YU, M., CRABB, D., XU, Y. & LIANGPUNSAKUL, S. 2011. Ethanol-induced alterations in fatty acid-related lipids in serum and tissues in mice. Alcohol Clin Exp Res, 35, 229-34.
SUBAUSTE, A. R., DAS, A. K., LI, X., ELLIOTT, B. G., EVANS, C., EL AZZOUNY, M., TREUTELAAR, M., ORAL, E., LEFF, T. & BURANT, C. F. 2012. Alterations in lipid signaling underlie lipodystrophy secondary to AGPAT2 mutations. Diabetes, 61, 2922-31.
ARIFIN, S. A. & FALASCA, M. 2016. Lysophosphatidylinositol Signalling and Metabolic Diseases. Metabolites, 6.
HAMMACK, B. N., FUNG, K. Y., HUNSUCKER, S. W., DUNCAN, M. W., BURGOON, M. P., OWENS, P. & GILDEN, D. H. 2004. Proteomic analysis of multiple sclerosis cerebrospinal fluid. Mult Scler, 10, 245-60.
TSUKAHARA, R. & UEDA, H. 2016. Myelin-related gene silencing mediated by LPA1 - Rho/ROCK signaling is correlated to acetylation of NFkappaB in S16 Schwann cells. J Pharmacol Sci,132, 162-165.
- introduction, lines 139-140: same for the abstract, ALS FRS-revised
Response: the term was revised.
Specific minor comments (abstract and introduction)
- line 21: development → for ALS is not proper, please prefer a more correct term. OK
- line 23: was → is; please use the present time for all the verbal expression referred to your research .OK
- line 34: ALS-FRS → version revised is more proper, ALS-FRSr. OK
Round 2
Reviewer 1 Report
Overall the response to comments is far from satisfactory. The review of the literature supporting one of the studies reveals a technical issue which raises doubts about the accuracy of the overall analysis.
For the units;
Response: Thanks for the comment. We decided to use only the raw data extracted from the articles in mg·dL-1 and not use unit conversions. Thus we have worked with the real published data without manipulating it. When we tried to do some units conversion, the values obtained were very different from those compared in the rest of the manuscripts. For this reason, we use the actual raw data that the authors provide us with in their publications. This adds a major inconvenience by greatly decreasing the number of values and scripts used. In addition, the units most commonly used in the clinic for serum lipid measurements are mg·dL-1.
This is not a very good rationale. Which the authors haven't discussed in the paper. Most of Europe uses mmol/L which is straightforward to accurately convert using available calculators or formula. The authors should convert values where these are available or exclude it using a different criteria.
- Table 4 has an issue, the value for the Ahmed study seems unlikely and I don't think its correct, that paper is also in mmol/L? Please correct this and corresponding numbers, correct discussion etc.
Response: Thanks for the comment. Data from Ahmed et al. were removed to calculate total cholesterol in the control cohort and the diseased cohort. Mean and error values already appear in the text excluding Ahmed's data (208.7 ± 1.7 and 206.2 ± 1.5 mg·dL-1, respectively). It was a mistake not to remove this data from Ahmed et al. in the table 4.
Although in the original article by Ahmed et al. indicates that the units are mg·dL-1, it was decided to eliminate them because these data were very different compared to those published in other publications. We thought it was more reliable to remove the total cholesterol data from the averages shown in Table 4.
This is an odd and incorrect response. I expected a thorough correction since this analysis had a major flaw. I read the Ahmed paper because the original article seemed to indicate a cholesterol concentration of 500mg/dl which is clearly an error. However, clearly the original paper uses mMol/L not mg/dl for all of these parameters. The exclusion of data would also need clarification. Also this data is still included in the forrest plot. This raises a serious doubt about the overall analysis, the haphazard nature of this correction does not fit with the underlying data.
Overall, unless there is a more basic referencing error, the paper could not be published because the incorrect analysis of the underlying data reflects a major insurmountable flaw.
Reviewer 2 Report
I think that the manuscript is now greatly improved.
However I have still other concerns:
Line 57. I think that the writing "contraceptive hormones" is not correct. Please check.
Lines 522-523. The sentence “These authors have even established the possible relationship between muscle atrophy and the taking of statins (atorvastatin) (drug used to lower serum cholesterol levels) [80].” Should be changed in: “These authors have even established the possible relationship between ALS, muscle atrophy and the taking of statins (atorvastatin) (drug used to lower serum cholesterol levels) [80].”
Round 3
Reviewer 1 Report
The authors response and modifications are satisfactory and the paper should be suitable for publication.